# CRISPRi-seq in *Haemophilus influenzae* reveals genome-wide and medium-specific growth determinants

Celia Gil-Campillo[1,2☺], Johann Mignolet[3☺], Asier Domínguez-San Pedro[1], Beatriz Rapún-Araiz[1,2], Axel B. Janssen[3], Vincent de Bakker[3], Jan-Willem Veening[3]*, Junkal Garmendia [1,2,4] *

1 Instituto de Agrobiotecnología, Consejo Superior de Investigaciones Científicas (IdAB-CSIC)-Gobierno de Navarra, Mutilva, Spain, 2 Centro de Investigación Biomédica en Red de Enfermedades Respiratorias (CIBERES), Madrid, Spain, 3 Department of Fundamental Microbiology, Faculty of Biology and Medicine, University of Lausanne, Lausanne, Switzerland, 4 Conexión Antimicrobial Resistance, Spanish National Research Council (AMR-CSIC), Madrid, Spain

☺ Equal contributions as first author.
* jan-willem.veening@unil.ch (J-WV); juncal.garmendia@csic.es (JG)

## Abstract

Work in the human pathobiont *Haemophilus influenzae* has pioneered functional genomics in bacteria such as genome-wide transposon mutagenesis combined with deep sequencing. These approaches unveiled a large set of likely essential genes, but functional studies are hampered due to a limited molecular toolbox. To bridge this gap, we engineered a titratable anhydrotetracycline-inducible CRISPRi (Clustered Regularly Interspaced Short Palindromic Repeats interference) platform for efficient regulation of gene expression in *H. influenzae*. Genome-wide fitness analyses in two different *in vitro* culture media by CRISPRi-seq revealed growth medium-dependent fitness cost for a panel of *H. influenzae* genes. We demonstrated that CRISPRi-programmed fitness defects can be rescuable, and we refined previous Tn-seq based essentialome studies. Finally, we introduce HaemoBrowse, an extensive user-friendly online resource for visual inspection of *H. influenzae* genome annotations, including sgRNA spacers. The inducible CRISPRi platform described here represents a valuable tool enabling functional genomics and the study of essential genes, thereby contributing to the identification of therapeutic targets for developing drugs and vaccines against *H. influenzae*.

## Author summary

CRISPRi-seq is a robust method to study bacterial gene fitness and essentiality via relative quantification and comparison of sgRNA abundance at a genome-wide scale. Here, we present a novel CRISPRi system for individual genes or

**Data availability statement:** Sequencing data are available as a BioProject, ID: PRJNA1274682.

**Funding:** C.G.-C. was funded by a PhD studentship from AEI, PRE2019-088382 and by a FEMS mobility grant. J.M. received funding from the European Union's Horizon 2020 research and innovation program (Marie Skłodowska-Curie grant N°101018461). A.D.-S.P. is funded by a PhD studentship from AEI, PRE2022-102925. A.B.J. was supported through a Postdoctoral Fellowship grant (TMPFP3_210202) from the Swiss National Science Foundation (SNSF), and a Robert Austrian Research Award from the International Society of Pneumonia and Pneumococcal Diseases. Work at J.W.V. laboratory was supported by the SNSF grants 310030_192517, 310030_200792 and NCCR 51NF40_180541. Work at J.G. laboratory was supported by grants from the Spanish AEI (PID2021-125947OB-I00 and PID2024-155918OB-I00), SEPAR (875/2019), Regional Govern of Navarra (PC150 and PC136), and from CIBER - an initiative from Instituto de Salud Carlos III (ISCIII), Madrid, Spain. The funders had no role in study design, data collection and analysis, decision to publish, or preparation of the manuscript.

**Competing interests:** I have read the journal's policy and the authors of this manuscript have the following competing interests: J.W.V. is a scientific advisory board member at i-Seq Biotechnology.

pooled libraries knockdown in *Haemophilus influenzae*. A genome-wide CRISPRi library designed to cover 99.27% of all total genetic features in the genome of RdKW20 strain was constructed and screened in two laboratory growth media through CRISPRi-seq, uncovering growth medium-dependent fitness cost, further confirmed with individual knockdown/knockout mutants. We also introduce HaemoBrowse (https://HaemoBrowse.VeeningLab.com), through which genome annotations and sgRNA design on *H. influenzae* genomes can be readily inspected. This platform provides a valuable tool for gene function and essentiality analyses in a notorious human pathobiont.

## Introduction

The human-specific Gram-negative bacterial species *Haemophilus influenzae* is a normal part of the upper airway microbiome but can cause invasive infections such as pneumonia, otitis media, and meningitis typically in infants and young children. Infant mortality due to invasive infections was drastically reduced by introduction of the conjugated vaccine that targets strains with the serotype b polysaccharide capsule [1]. However, nontypeable *H. influenzae* strains continue to cause high morbidity due to their role in common infections and chronic diseases, and are major contributors to persistent infection and exacerbations of chronic obstructive pulmonary disease (COPD) and cystic fibrosis (CF) [2–4].

Functional genomics is a powerful approach to identify new vaccine candidates and antibiotic targets for important human pathogens. Genome-wide studies to assess the fitness contribution or essentiality of each genetic feature across diverse conditions or genetic backgrounds in bacteria have predominantly been performed using transposon-mediated approaches combined with next-generation sequencing (e.g., Tn-seq and TraDIS) [5]. Indeed, Tn-seq has identified putative essential genes for *H. influenzae* adaptation to changes in environmental $CO_2$ levels, for survival in the presence of neutrophils, serum and complement, and for *in vivo* survival in a murine model of airway infection [6–9]. However, transposon mutagenesis methods are less appropriate for gene essentiality studies as insertion mutants of essential genes are per definition counter-selected. In contrast, inducible Clustered Regularly Interspaced Short Palindromic Repeats interference (CRISPRi) technology, developed from natural CRISPR systems found in bacteria and archaea as part of their adaptive immunity against phages, overcomes this limitation as it enables the transient repression of target genes and allows to monitor phenotypes associated to dispensable and essential genes [10–12]. CRISPRi is based on the co-expression of a single-guide RNA (sgRNA) and a catalytically inactive (dead) *Streptococcus pyogenes* Cas9 (dCas9) protein, which binds DNA but lacks endonuclease activity. The sgRNA consists of a 20 nucleotide-long variable spacer sequence that is complementary to a target DNA stretch, a Cas9 handle crucial for interaction with dCas9, and a transcriptional terminator. The sgRNA targets the dCas9 protein to a site on the genome with a sequence complementary to its spacer sequence next to a

protospacer adjacent motif (PAM) site composed of 3 nucleotides, typically NGG. When bound to the non-template strand of the target gene, the dCas9•sgRNA complex serves as a roadblock for RNA polymerase, hampering transcription initiation or elongation of the targeted gene [10–15]. sgRNAs can be designed to target any sequence of interest provided that the target is next to a PAM site. As such, almost all genes of a given genome can be studied with an appropriate design of sgRNAs if dCas9 and/or the sgRNA is under the control of an inducible promoter to prevent constitutive gene repression. This makes CRISPRi particularly valuable for functional studies including essential gene assessment [16–19].

Two approaches can be employed for genome-wide CRISPRi screenings. An arrayed library method involves handling each mutant with a single sgRNA individually, enabling direct observation of diverse phenotypes [15,17,20]. This method is laborious and time consuming considering each sgRNA mutant strain needs individual processing. In contrast, a pooled library approach, where multiple knockdown strains are grown together, is more convenient but it limits phenotype measurements besides gene fitness. Deep sequencing reveals changes in sgRNA abundances of the pooled library upon selective pressures, reflecting the differential fitness provoked by conditional target repression. Hence, the combination of pooled CRISPRi libraries with NGS (CRISPRi-seq) offers a robust method to study gene fitness in bacteria (via relative quantification and comparison of sgRNA abundance) at a genome-wide scale in particular growth conditions [16,18–27].

Here, we implemented a CRISPRi system suitable for individual genes or pooled libraries knockdown in *H. influenzae*. We first validated operability and titratability of the newly implemented anhydrotetracycline (aTc)-inducible CRISPRi system. Furthermore, we constructed a genome-wide CRISPRi library with high coverage in the genome of RdKW20 strain, designed to cover 99.27% of all total genetic features. As a proof of concept, the generated *H. influenzae* genome-wide CRISPRi library was screened in two laboratory growth media to quantify gene fitness through CRISPRi-seq. We uncovered growth medium-dependent fitness cost, and confirmed several hits with individual knockdown/knockout mutants. In addition, we introduce HaemoBrowse (https://HaemoBrowse.VeeningLab.com), through which the genome annotations and sgRNA design on *H. influenzae* genomes can be readily inspected. The *H. influenzae* CRISPRi platform presented here provides a valuable tool for both gene functional analyses and essentiality in a notorious human pathobiont.

## Results

### An inducible CRISPRi system in *H. influenzae*

In previous work, we successfully established CRISPRi for the Gram-positive human pathogen *Streptococcus pneumoniae* by placing the *dcas9* gene under tight control of LacI and TetR-regulated promoters, while expressing the sgRNA from the synthetic constitutive P3 promoter [17,19]. To adapt this operational streptococcal CRISPRi system for *H. influenzae*, we integrated two cassettes into the chromosome of the widely used laboratory strain *H. influenzae* RdKW20, by taking advantage of its high frequency to naturally acquire exogenous linear DNA. The first cassette was engineered as a linear fragment through Golden Gate cloning encoding *dcas9* under the control of the anhydrotetracycline (aTc)-responsive promoter (hereafter P$_{tet}$) [28], an *erm* erythromycin resistance gene, and the *tetR* transcriptional repressor. The product was integrated into the non-essential *xylB-rfaD* locus of the RdKW20 chromosome [29], generating the RdKW20-*dcas9* strain (hereafter, *dcas9* strain) (Fig 1A). This *dcas9* cassette was designed to have no- or minimal impact on the neighbour genes. First, the *xylB* gene is upstream of- and in the same orientation than the *dcas9* cassette. Second, the *dcas9* cassette is bordered by transcriptional terminators to minimize transcriptional readthrought. Third, neither deletion nor duplication were generated due to the *dcas9* cassette chromosomal integration. Integration of the cassette at this locus did not hamper the fitness of the engineered strain compared to its isogenic parental wild-type (**Fig A in S1 Text, panel A**). In parallel, we constructed the pPEPZHi-*mCherry* vector, which will be used to constitutively express sgRNAs from the P$_3$ promoter [14]. pPEPZHi-*mCherry* encodes the *mCherry* gene to allow for visual red-white screening of sgRNA cloning in *E. coli*, the dCas9-required handle sequence, and a *spec* spectinomycin resistance gene for selection in both *Escherichia coli* and *H. influenzae*. Illumina read 1 and read 2 sequences flank the sgRNA sequence to allow for a one-step PCR amplification of the sgRNAs and downstream deep sequencing of a library pool. Finally, we introduced flanking

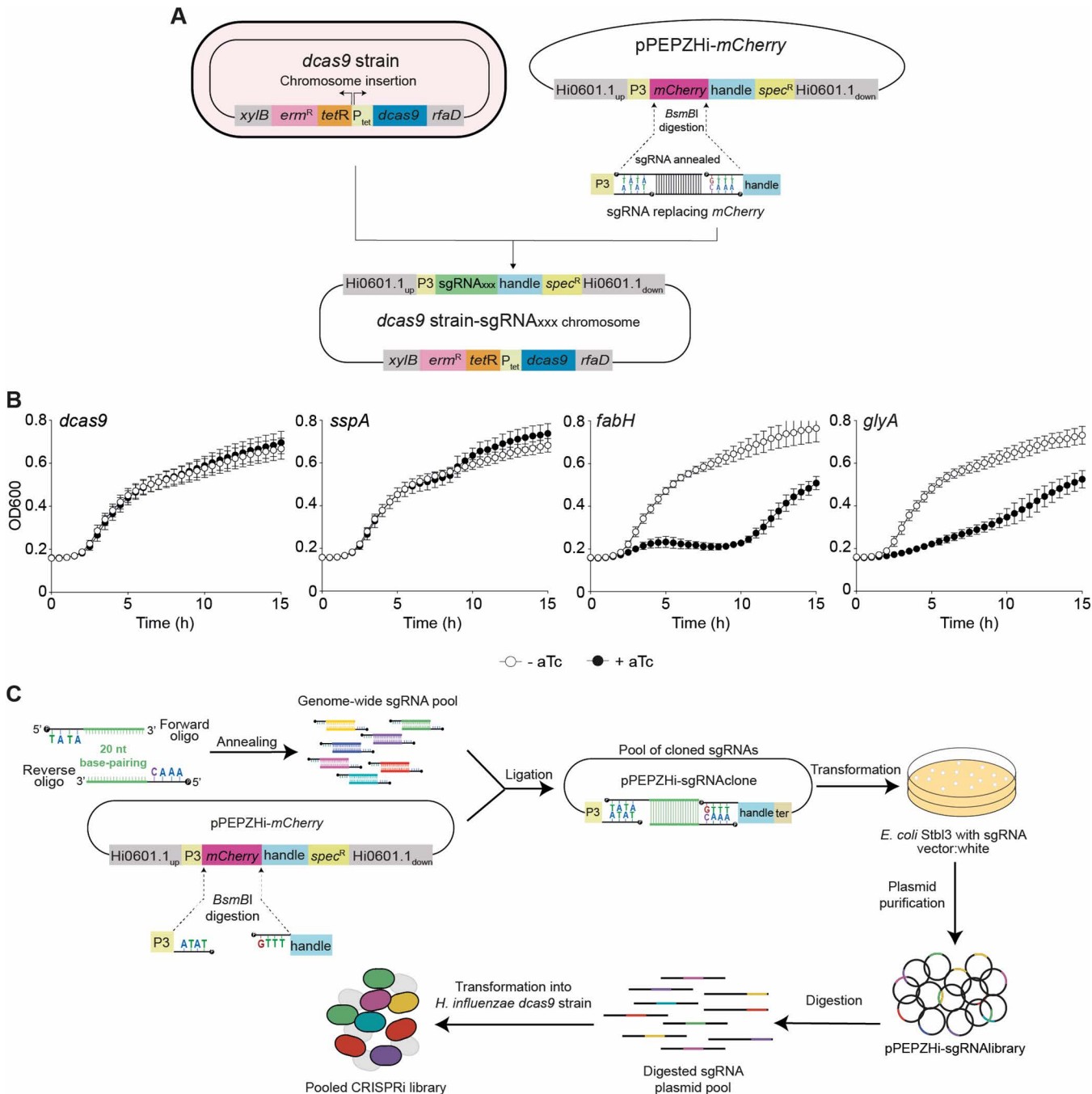

**Fig 1. Engineering of a CRISPRi knockdown platform for *H. influenzae*.** (A) Generation of a *H. influenzae* strain with an aTc-inducible *dcas9* by double homologous recombination at the *xylB/rfaD* locus. Backbone vector pPEPZHi-*mCherry* for sgRNA cloning includes a Spec[R] marker, a P3 promoter controlling sgRNA expression, and an *mCherry* gene encoding a red fluorescent protein flanked by *BsmB*I restriction sites. *BsmB*I vector digestion generates two ends compatible for annealed sgRNA cloning by *mCherry* gene replacement. The sgRNA cassette is amplified from pPEPZHi-sgRNA$_{xxx}$ to perform chromosomal integration at the HI0601.1 *locus* of the *dcas9* background. (B) Functional validation of the *H. influenzae* CRISPRi genetic platform. The *dcas9* strain (-) and derivatives expressing *sspA*, *fabH* or *glyA* sgRNAs were grown in sBHI, in the absence (white circles)/presence (black circles) of aTc for CRISPRi-based gene silencing. Growth was monitored by measuring OD$_{600}$ every 30 min for 15 h; standard deviation to the mean is shown for each timepoint. (C) Workflow to generate a CRISPRi-based genome-wide library in *H. influenzae*. Oligo pairs containing 20 bp complementary stretchs and 4 nt overhangs compatible with the *BsmB*I-digested pPEPZHi-*mCherry* vector were annealed, phosphorylated and ligated in pool to the *BsmB*I-digested pPEPZHi-*mCherry*. The pool of purified sgRNA-containing plasmids (pPEPZHi-sgRNAlibrary) serves as the reservoir for CRISPRi

library construction in *H. influenzae*. The sgRNA library plasmid pool was linearized and transformed into the competent *dcas9* background, generating the pooled CRISPRi library. Both *dcas9* and sgRNA cassettes are integrated in two separate locations into the recipient host chromosome, *xylD/rfaD* and HI0601.1 loci, respectively.

homology regions for the neutral Hi0601.1 locus [30], to subsequently generate sgRNA-containing linear cassettes for chromosomal integration in the *dcas9* strain (double homologous recombination) by natural transformation (Fig 1A).

To assess the efficacy of our CRISPRi system in *H. influenzae*, we separately integrated three sgRNAs targeting either essential or non-essential (control) genes [31–33] in the chromosome of the *dcas9* background strain. As shown in Fig 1B, strains carrying a sgRNA targeting *fabH* (encoding a β-ketoacyl-acyl carrier protein synthase III) or *glyA* (encoding a serine hydroxymethyltransferase) were severely hampered to grow upon addition of 50 ng/mL aTc, in line with their pre-dicted essential functions in the cell. In contrast, the strain with a sgRNA targeting the dispensable gene *sspA* (encoding stringent starvation protein A) grew similarly whether dCas9 was induced or not. As a control, we verified that expression of dCas9 had no effect on growth in a sgRNA-devoid strain (Fig 1B, left panel).

For more subtle control over gene expression, and to determine the dynamic range of the CRISPRi system in *H. influenzae*, we tested the growth profiles for a range of aTc concentrations. We observed a narrow window of gene expression control between 0.25 and 1 ng/mL aTc indicating that the here-used $P_{tet}$ promoter offers some levels of titratability in dCas9 expression (**Fig A in** S1 Text**, panel B**). For the rest of the work presented here, we used a saturating aTc concentration of 50 ng/mL.

The results presented demonstrated that the developed CRISPRi system for *H. influenzae* is both functional and induc-ible. To further verify the accuracy of the system, we selected *ftsZ* and *dnaA* as knockdown targets. Since these targets are directly involved in cell division and DNA replication initiation, respectively, we can evaluate the knockdown pheno-type by observing bacterial cell and nucleoid morphology. We engineered sgRNA$_{ftsZ}$ and sgRNA$_{dnaA}$ strains as previously described. As expected, cell growth was inhibited for both depletion strains upon dCas9 induction (Fig 2A). This macro-scopic behavior correlates with an elongation phenotype of most bacterial cells that does not occur when using a sgRNA targeting a neutral gene (*sspA*) (Fig 2B). As highlighted by DAPI staining, activation of the CRISPRi system revealed two different phenotypes for *ftsZ* and *dnaA* respective gene knockdowns. Indeed, *ftsZ* knockdown produced filamentous cells with multiple compartmentalized nucleoids or nucleoids that spread throughout the cytoplasm, while cells depleted for *dnaA* harbored a unique compact nucleoid with DNA-free area and produced anucleate minicells (Fig 2C). These obser-vations are in direct line with the functions performed by FtsZ and DnaA, demonstrating that the here-described CRISPRi system for *H. influenzae* is specific.

## Development of a genome-wide *H. influenzae* CRISPRi library

To widen the scope of this new tool in *H. influenzae*, we developed a pooled library for strain RdKW20. Through previously described and validated computational algorithms [19], we designed an optimized library including 1,773 sgRNAs target-ing all predicted genetic features annotated in the RdKW20 genome. This library covers 99.27% of all total features of the RdKW20 genome (13 features are not targeted by the library due to the lack of PAM sites in their coding sequences). Of note, some designed sgRNAs may target more than one genetic feature (coding sequences, ribosomal RNAs, transfer RNAs or small RNAs) if that feature is present in multiple copies, or if repetitive regions are present (S1 Dataset).

To construct the CRISPRi sgRNA library, we used a synthetic pool of 3,546 ssDNA oligonucleotides for RdKW20 encoding two partially complementary oligonucleotides for each sgRNA (1,773 sgRNAs). From this synthetic ssDNA pool, a pooled plasmid library was constructed through cloning into *BsmB*I-digested pPEPZHi-*mCherry* and transformation into *E. coli* Stbl3 (Fig 1C). In total, the cloning yielded ~$10^{10}$ CFU/mL, ~$4.48 \times 10^6$ times the theoretical coverage of the library, of which ~0.1% was positive for mCherry, indicating a highly efficient digestion and cloning reaction. Subsequent Illumina

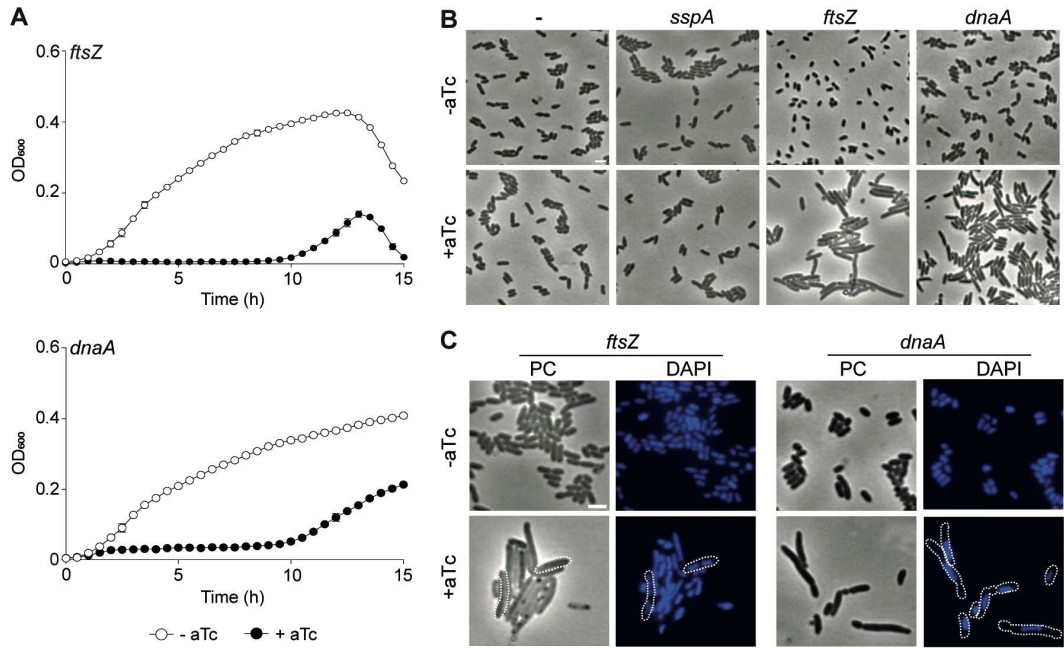

**Fig 2. Efficiency and specificity of the *H. influenzae* CRISPRi system.** (A) The *dcas9* derivative strains expressing *ftsZ* or *dnaA* sgRNAs were grown in sBHI, in the absence (white circles)/presence (black circles) of aTc, in 96-well plates; OD$_{600}$ was measured every 30 min for 15 h. Standard deviation to the mean is shown for each timepoint. (B) Phase contrast (PC) microscopy showing bacterial cell morphology for the *dcas9* strains (-) and derivatives expressing *sspA*, *ftsZ* or *dnaA* sgRNAs. Bacteria were grown in sBHI in the absence/presence of aTc. (C) Detailed bacterial cell and nucleoid morphology alterations showing the effect of dCas9 induction in *dcas9* derivative strains expressing *ftsZ* or *dnaA* sgRNAs. The nucleoid was stained with DAPI and several cells are outlined to emphasize nucleoid structures and organization in the cytoplasm. Scale bars are equal to 2 μm.

sequencing of the plasmid library showed a normal distribution of the relative sgRNA prevalence, with only few sgRNAs absent [16] in the pool (**Fig B in** S1 Text**, panel A**, and S2 Dataset). The created plasmid library was subsequently used for digestion (see Methods section) and natural transformation of the sgRNA-containing linear cassette pool into the P$_{tet}$-*dcas9* background strain, resulting in ~1x10$^6$ total CFU transformants, and retention of 1,742 sgRNA in a normally distributed library (98.3% of all 1,773 synthesized sgRNAs; 1,701 genes (HI_XXX) targeted by sgRNAs out of 1,779) (Fig 1C and **Fig B in** S1 Text**, panel B**, and S2 Dataset).

In summary, we were able to generate a CRISPRi library in *H. influenzae* that includes evenly distributed sgRNAs targeting 98.3% of all annotated features.

### Genome-wide *H. influenzae* fitness evaluation by CRISPRi-seq

Next, we used the CRISPRi library to investigate the RdKW20 strain 'essentialome', by quantifying the fitness of each sgRNA target when grown in sBHI medium. For a better resolution of CRISPRi-seq across all sgRNAs, three time points (7, 14 and 21 generations) were sampled in quadruplicate (Fig 3A). Induction of dCas9 was achieved by adding aTc at 50 ng/mL.

Through deep sequencing, quantification and comparison of sgRNA abundance in induced *versus* uninduced samples, we confirmed the tight control over *dcas9* expression by P$_{tet}$, as all uninduced samples had a similar sgRNA composition, whether they were grown for 7, 14, or 21 generations (**Fig C in** S1 Text**, panels A** and **B**). Considering induced samples, the majority (91%) of variance between samples was due to *dcas9* induction in a time-dependent manner. Fitness score comparison between the three conditions showed that the number of essential genes gradually increased according to the

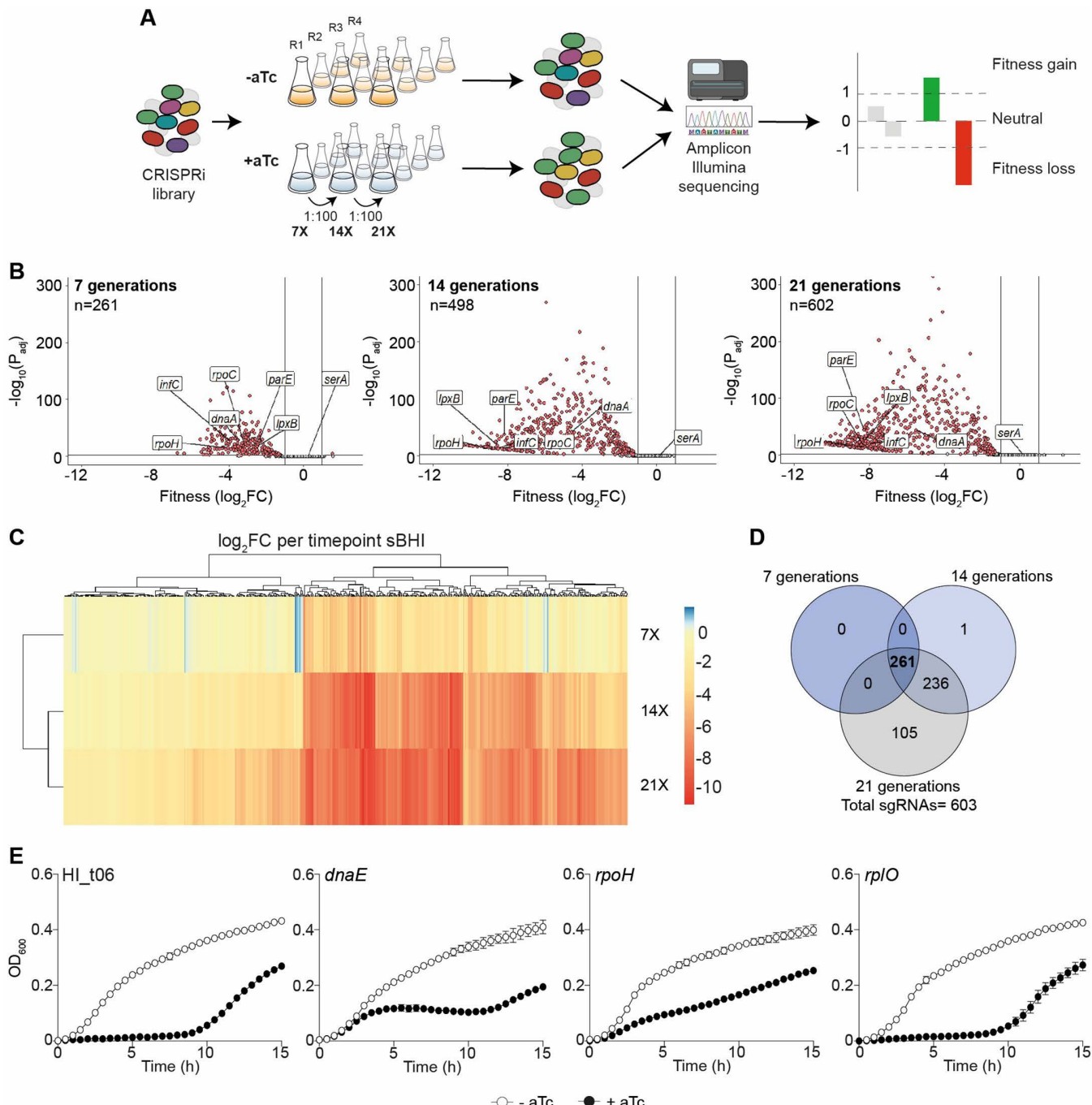

**Fig 3. CRISPRi-seq gene fitness analysis of *H. influenzae* during growth in sBHI medium.** (A) The *dcas9* CRISPRi library was grown in sBHI without (orange) or with (blue) aTc 50 ng/mL. Four biological replicates were performed for each condition. After 7, 14 and 21 generations of exponential growth, bacterial cultures were pelleted, genomic DNA was purified, sgRNA-containing amplicons were generated and Illumina sequenced, and data were analysed for differential fitness score abundance. (B) Volcano plots show sgRNA target fitness scores upon *dcas9* induction for 7, 14 and 21 generations of exponential growth in sBHI. Red dots represent sgRNAs targeting essential genes, with significant differential fitness effects ($log_2FC < -1$ and $P_{adj} < 0.05$; vertical and horizontal black lines). Several essential (*dnaA* [DNA replication initiation], *infC* [translation initiation factor], *lpxB* [lipid A synthase], *parE* [DNA topoisomerase], *rpoC* [RNA polymerase] and *rpoH* [sigma factor]) and dispensable (*serA* [serine metabolism]) genes in sBHI have been plotted. (C) The heatmap combined with hierarchical clustering displays all significantly enriched/depleted sgRNAs over the three timepoints. A negative (red) score indicates a fitness loss in inducing conditions, while a positive (blue) score indicates a fitness gain. (D) Venn diagram showing the

overlap of sgRNAs with reduced abundance after aTc induction over the three timepoints. (E) CRISPRi-based analysis of *H. influenzae* specific gene fitness. *dcas9* derivative strains expressing HI_t06, *dnaE*, *rpoH* or *rplO* sgRNAs were grown in sBHI, in the absence (white circles)/presence (black circles) of aTc. Strains were grown in 96-well plates, and $OD_{600}$ was measured every 30 min for 15 h. Standard deviation to the mean is shown for each timepoint.

number of generations (Fig 3B-D, and S3 Dataset). Indeed, within the induced samples, we observed that 261 sgRNAs have significantly reduced abundance after 7 generations, 498 after 14 generations and 602 after 21 generations (Fig 3B and 3D, and S3 Dataset). Consistently, sgRNAs with significant lower abundances at earlier timepoints remain so at later timepoints (Fig 3D). Moreover, volcano plots and heatmaps highlight the trend for individual fitness scores to be lower and lower over generations (Fig 3B and 3C, and **Fig C in** S1 Text**, panel B**). In direct line with individual sgRNA data (Figs 1B and 2), sgRNAs targeting *glyA, dnaA, fabH* or *ftsZ* showed drastic reduced abundances (S3 Dataset).

To validate this CRISPRi-seq fitness screen performed in sBHI growth medium, we independently analyzed a panel of genes whose sgRNAs showed significantly reduced abundance after 7 generations. Fourteen separate sgRNAs targeting the HI_t06, *dnaE*, *rpoH*, *rplO*, *infC*, *rpsL*, *ispE*, *metK*, *rpoC*, *lpxB*, *parE*, *rsxA*, *rpsT* and *mepA* genes were individually cloned in the pPEPZHi-*mCherry* vector and integrated in the $P_{tet}$-*dcas9* background, as described above. In the presence of inducer, twelve out of the fourteen tested sgRNAs strains exhibited growth defects (Fig 3E and **Fig D in** S1 Text).

These results strongly validate the robustness and reliability of this novel *H. influenzae* CRISPRi platform in bacterial genome-wide assays, and unveil gene requirements in a laboratory-based complex medium.

### *H. influenzae* CRISPRi-seq reveals growth medium-dependent essentiality

We next assessed *H. influenzae* gene essentiality in a different growth condition through cultivation in a chemically defined (CDM) medium [34,35], and subsequent comparative analysis to the sBHI data. The CRISPRi library was grown in CDM in the absence/presence of aTc, and deep-sequenced for gene fitness quantification. Akin to sBHI, variability was primarily attributed to *dcas9* expression with longer induction times increasing differences in sgRNA content (Fig C in S1 Text, panels A and B), as the number of genes with significant fitness defects were 259 sgRNAs with reduced abundance after 7; 513 after 14; and 590 after 21 generations (Fig 4A-C, and S3 Dataset). As reported hereabove for the sBHI screen, we validated the functionality of this high-throughput CRISPRi screen in CDM with 12 (out of 14) individual sgRNA strains (Fig 4D and Fig E in S1 Text).

A significant number of essential genes were common to both sBHI and CDM media (531 genes). As expected, analysis of Clusters of Orthologous Genes (COG) for all essential genes (80.3% shared essential genes between the two media) showed minor differences. Translation, ribosomal structure and biogenesis was the most affected category. Small differences between the two media were observed for inorganic ion transport, nucleotide and amino acid metabolism and transport, as well as energy production (Fig 5A).

Next, we performed a differential fitness analysis between samples grown in sBHI *versus* CDM. Comparative fitness analysis showed that 7, 35 and 60 genes were significantly more essential in sBHI at 7, 14, and 21 generations, respectively, whilst 22, 75 and 94 genes were significantly more essential in CDM (Fig 5B, and S4 and S5 Datasets). Through a KEGG pathway enrichment analysis on the genes differentially essential after 21 generations, we observed that more essential genes in CDM mostly belong to amino acid and nucleotide biosynthesis pathways (36% and 17%, respectively) and metabolic pathways in a more general manner (85%) (Fig 5C). In contrast, sBHI-specific essential genes primarily encode transporters (35%) (**Fig F in** S1 Text). Aiming to confirm this medium-dependent essentiality, we selected 2 genes, *guaB* and *ilvE*, whose sgRNAs showed reduced abundance in CDM but stayed equal in sBHI. The *guaB* gene encodes an inosine-5´-monophosphate dehydrogenase that catalyzes the conversion of inosine 5'-phosphate to xanthosine 5'-phosphate, the first committed and rate-limiting step in the *de novo* synthesis of guanine nucleotides. The *ilvE* gene encodes for a branched chain amino acid aminotransferase acting on leucine, isoleucine and valine. Both sgRNAs were individually cloned in the $P_{tet}$-*dcas9* background and bacterial growth was assayed in CDM and sBHI (in the absence/

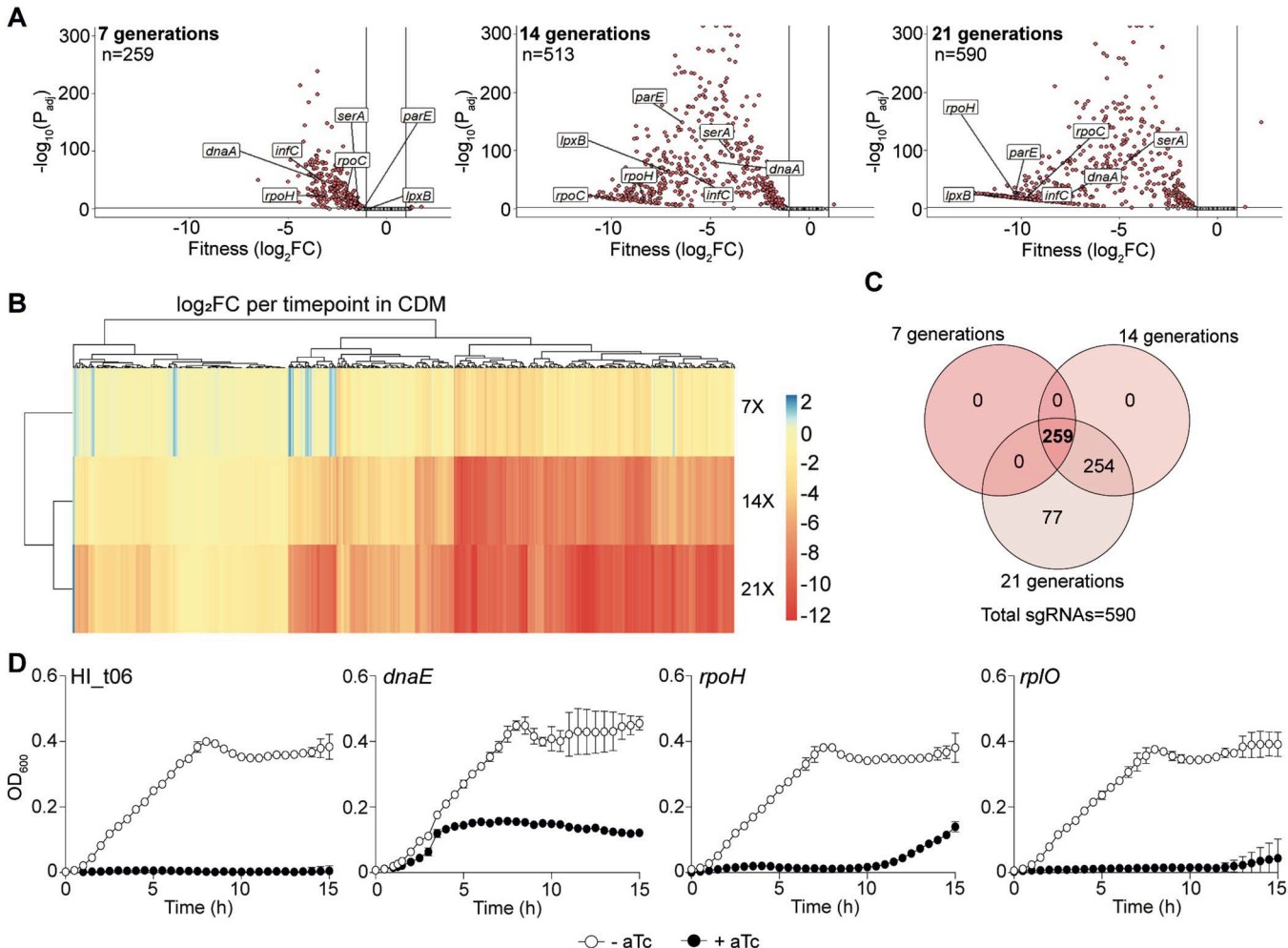

**Fig 4. CRISPRi-seq gene fitness analysis of *H. influenzae* during growth in CDM medium.** (A) Volcano plots show sgRNA target fitness scores upon *dcas9* induction for 7, 14 and 21 generations of exponential growth in CDM. Red dots represent sgRNAs targeting essential genes, with significant differential fitness effects (log$_2$FC<−1 and P$_{adj}$<0.05; vertical and horizontal black lines). Several essential genes (*dnaA* [DNA replication initiation], *infC* [translation initiation factor], *lpxB* [lipid A synthase], *parE* [DNA topoisomerase], *rpoC* [RNA polymerase], *rpoH* [sigma factor] and *serA* [serine metabolism]) in CDM have been plotted. (B) The heatmap combined with hierarchical clustering displays all significantly enriched/depleted sgRNAs over the three timepoints. A negative (red) score indicates at fitness loss in inducing conditions, while a positive (blue) score indicates a fitness gain. (C) Venn diagram showing the overlap of sgRNAs with reduced abundance after induction over the three timepoints. (D) *dcas9* derivative strains expressing HI_t06, *dnaE*, *rpoH* or *rplO* sgRNAs were grown in CDM, in the absence (white circles)/presence (black circles) of aTc. Strains were grown in 96-well plates, OD$_{600}$ was measured every 30 min for 15 h. Standard deviation to the mean is shown for each timepoint.

presence of aTc). Strikingly, dCas9 induction showed a CDM-specific growth defect, while growth in sBHI was unaffected regardless of dCas9 induction (Fig 5D and 5E).

These results validate the use of this CRISPRi platform in bacterial genome-wide assays to reveal growth condition-dependent gene requirements.

## CRISPRi-programmed fitness defects are rescuable in *H. influenzae*

Our differential fitness analysis between sBHI or CDM grown samples shed light on medium specificities and genetic targets (**Figs G** and **H in** S1 Text) [31,36]. Among others, genes encoding enzymes required for threonine and serine

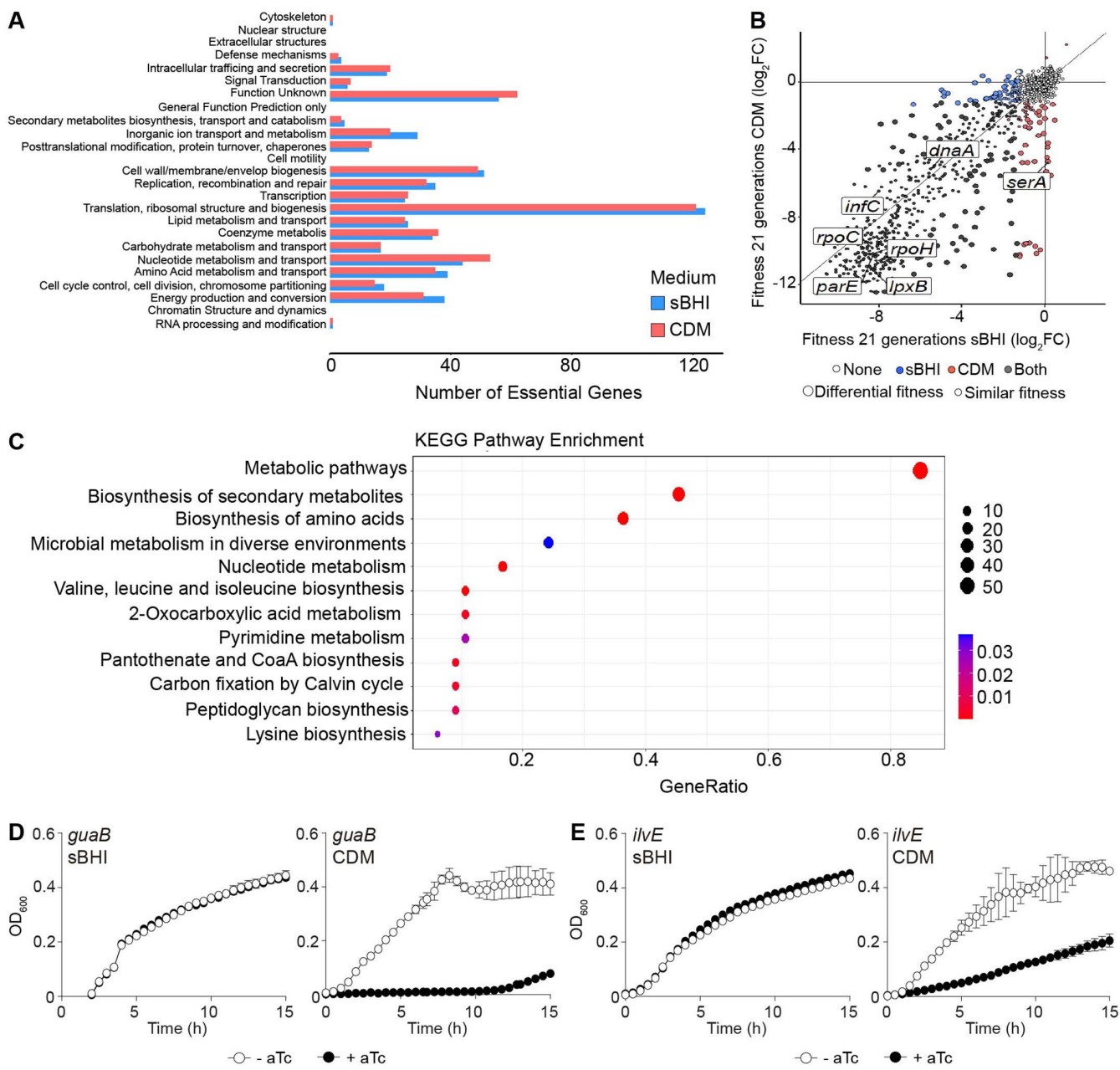

**Fig 5. CRISPRi-seq reveals medium-specific growth determinants for *H. influenzae*.** (A) Distribution of COG categories in all significantly essential genes in sBHI (blue) and CDM (red). (B) Interaction plot comparing fitness scores upon *dcas9* induction for 21 generations in sBHI *versus* CDM. Blue and red dots represent sgRNAs with $\log_2 FC < -1$ and $P_{adj} < 0.05$ in sBHI and CDM, respectively. Black dots represent sgRNAs with $\log_2 FC < -1$ and $P_{adj} < 0.05$ in both conditions, and white dots color dispensable genes. Dot size refers to genes considered as differentially essential in one medium (94 genes more essential in CDM; 60 genes more essential in sBHI). Several genes (*dnaA* [DNA replication initiation], *infC* [translation initiation factor], *lpxB* [lipid A synthase], *parE* [DNA topoisomerase], *rpoC* [RNA polymerase], *rpoH* [sigma factor] and *serA* [serine metabolism]) essential in at least one medium are labeled. (C) KEGG pathway enrichment analysis in CDM (*versus* sBHI). The gene ratio refers to the number of enriched genes in CDM for a specific pathway reported to the total amount of differentially essential genes. Dot size reflects the number of differentially essential genes. Color shades scale the significance ($P_{adj}$). (D & E) *dcas9* derivative strains expressing sgRNA$_{guaB}$ (D) or sgRNA$_{ilvE}$ (E) were grown in sBHI and CDM, in the absence (white circles)/presence (black circles) of aTc. Strains were grown in 96-well plates, OD$_{600}$ was measured every 30 min for 15 h, standard deviation to the mean is shown for each timepoint.

biosynthesis are differentially essential in CDM compared to sBHI (Fig 6 and **Fig I in** S1 Text). As a last validation of this *H. influenzae* CRISPRi system, we tested whether the observed CRISPRi-promoted fitness burden could be rescued by other genetic tools. We selected the CDM-specific *serA* gene, which encodes a D-3-phosphoglycerate dehydrogenase involved in the L-serine biosynthesis pathway. Therefore, repression of the *serA* gene should cause L-serine auxotrophy. Coherently, in the presence of aTc, bacterial growth of the sgRNA$_{serA}$ strain exhibited a severe defect in CDM, but not in sBHI (Fig 6A). In line with this data, a Δ*serA* deletion mutant showed auxotrophy when grown in CDM (Fig 6B). L-serine concentration is about 285 mM in sBHI (Javier Asensio-López, personal communication), while only 0.29 mM in CDM [35].

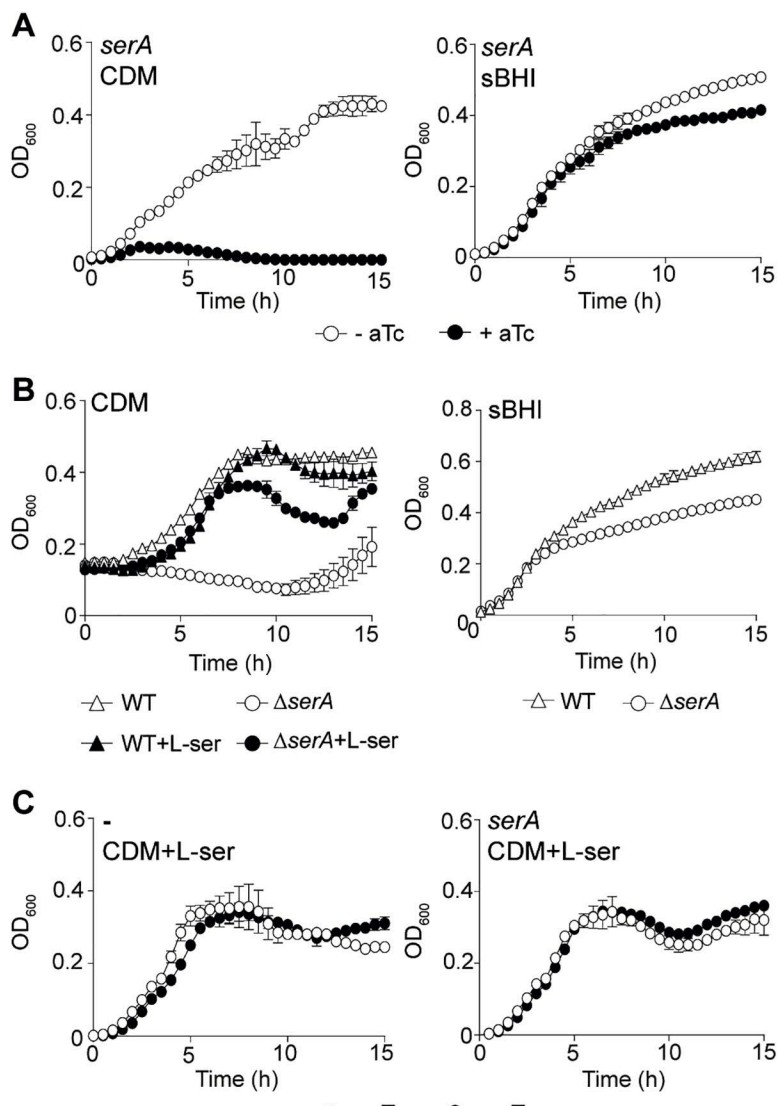

**Fig 6. sgRNA gene silencing reversibility in *H. influenzae*.** (A) The *dcas9* derivative strain expressing sgRNA$_{serA}$ was grown in CDM (left panel) and sBHI (right panel), in the absence (white circles)/presence (black circles) of aTc. (B) RdKW20 WT (triangles) and Δ*serA* (circles) strains were grown in CDM in the absence (white) or presence (black) of L-serine 10 mM (left panel), and in sBHI (right panel). (C) *dcas9* strain and its derivative expressing sgRNA$_{serA}$ were grown in CDM supplemented with L-serine 10 mM, in the absence (white circles)/presence (black circles) of aTc. Strains were grown in 96-well plates, OD$_{600}$ was measured every 30 min for 15 h, standard deviation to the mean is shown for each timepoint.

We reasoned that increasing L-serine concentration in CDM might abolish the sgRNA$_{serA}$ and ΔserA fitness defects. As expected, supplementation of CDM with L-serine 10 mM abolished the growth arrest phenotype caused by underexpression of serA (Fig 6B and 6C). For completion, similar observations were made for the dcas9-sgRNA$_{serA}$ and ΔserA strains when using a minimal chemically defined medium totally deprived of L-serine (mCDM) (**Fig J in** S1 Text). These results fully support the operativity of our study design and CRISPRi platform to identify H. influenzae genes undergoing medium-dependent fitness cost.

### HaemoBrowse: a visual and intuitive *H. influenzae* genome browser

Finally, we developed HaemoBrowse, a genome browser for *H. influenzae* (Fig 7). Based on JBrowse 2, HaemoBrowse offers a user-friendly and highly intuitive visual platform to browse the genome of the strain RdKW20 for its encoded features. Moreover, HaemoBrowse shows the position and sequence of the designed sgRNAs. Additional flexibility is offered by allowing for the search of locus tags (including HI_XXXX, locus tags for RdKW20) and gene names. We also included the genomes of three other commonly used *H. influenzae* strains, NTHi375, R2866 and 86–028NP. To allow for further exploration of gene function, direct links to the UniProt [37], PaperBLAST [38], AlphaFold [39], FoldSeek [40], and STRING databases [41] are available for each encoded feature. Finally, the conservation and the essentiality of the encoded features in the different conditions and at different timepoints, as determined through the CRISPRi-system introduced in RdKW20 here, are documented. HaemoBrowse is freely available at https://HaemoBrowse.VeeningLab.com.

## Discussion

Identification of bacterial genetic targets for drug development requires powerful screening strategies and extended information on gene function and bacterial physiology to lead such pharmacological search. Here, we made a step forward towards this goal for *H. influenzae* by designing a robust genome-wide screen. We engineered a scalable CRISPRi platform and developed a CRISPRi library covering 99.27% of all annotated features in the genome of the *H. influenzae* RdKW20 strain. We demonstrate the functionality of our CRISPRi system at a macroscopic and microscopic scale in the RdKW20 strain, and employed CRISPRi-seq to quantify bacterial gene fitness *in vitro* on a genome-wide level. Moreover, given that this CRISPRi system is responsive to any type of tetracycline-like molecule including doxycycline, challenging vertebrate/murine models with an induced/uninduced pool of our CRISPRi library is likely to identify genes involved in virulence, invasion or colonization at multiple steps of infection, as demonstrated for *S. pneumoniae* [19].

Our high-throughput knockdown approach is validated by a previous transposon-based knockout essentialome study in sBHI [42]. Even more, 83.7% (262 out of 313) of the essential genes identified via Tn-seq across three independent studies (*in vitro* and *in vivo*, i.e., in a mouse intranasal infection model) [9,32,42] were also identified at 21 generations in sBHI. Besides, 340 genes were found to be only essential when using our CRISPRi-seq approach, and 47 genes were found to be only essential in Tn-seq-based previous analyses (S6 Dataset).

Importantly, the inducibility of our system allowed us to refine the study of essential genes. For instance, 47 genes that are significantly essential in both CDM and sBHI are considered as more essential in CDM, providing an exhaustive view of the genetic requirements for bacterial growth in this specific medium. This kind of differential analysis on crucial genes is obviously precluded with Tn libraries where clones bearing a Tn insertion in an essential gene are counter-selected and eradicated from the pool of Tn mutants. Also, sampling at multiple time points provided a dynamic view of *H. influenzae* genes important for fitness. In a logical way, a prolonged inducer treatment aggravates fitness losses, resulting in larger fold changes. This may also help to identify genes with marginal fitness effects over generations. In addition, this temporal analysis gives information about the kinetics of depletion that, besides the efficiency of sgRNA repression, reflects the necessity, abundancy and stability of the encoded protein.

Of note, a decrease in the abundance of an sgRNA does not automatically indicate that the gene is essential as such, but rather statistically less abundant than in the uninduced control under a given condition. Therefore, CRISPRi data

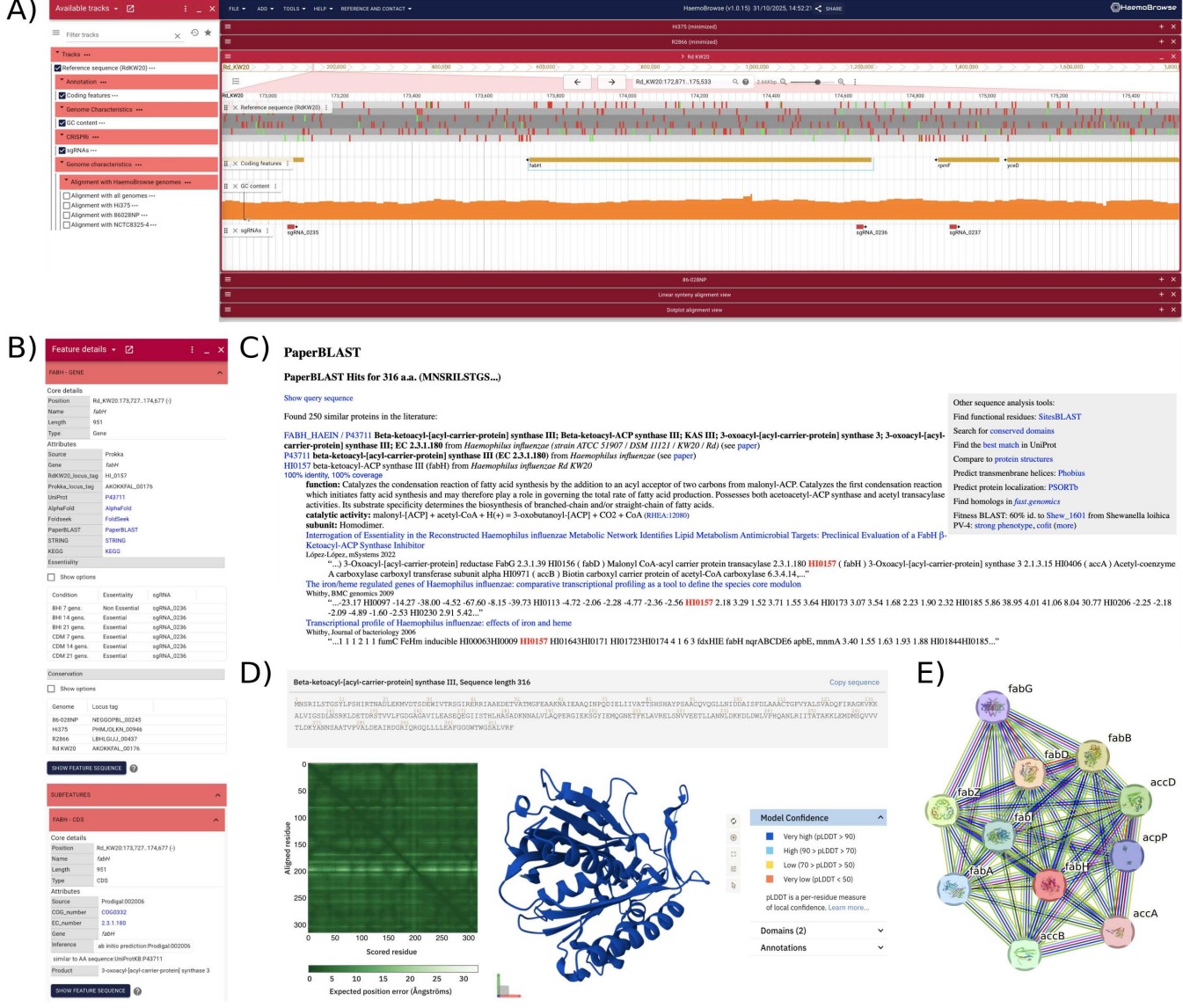

**Fig 7. HaemoBrowse, available at https://HaemoBrowse.VeeningLab.com.** (A) A screenshot of the *fabH locus* as shown in HaemoBrowse. In the left panel, tracks can be turned on/off. In the right panel, the genome can be browsed by dragging the mouse to the left or right, zooming in and out, or searched on gene name and/or *locus* tags. (B) For each coding sequence, a context menu provides additional information and links to external resources, such as Uniprot [37], PaperBLAST [38], AlphaFold [39], FoldSeek [40], and STRING [41]. The context menu also displays the essentiality per experimental condition. (C) PaperBLAST provides additional information through literature search [38]. (D) Structure predictions can be consulted through AlphaFold [39]. (E) STRING database shows interactions of the protein of interest [41].

interpretation strongly requires considering the robustness of the significance. In addition, CRISPRi is by nature polar thus multiple genes in polycistronic operons can be affected by a single sgRNA. It is thus always advised to consider the genetic context of the CRISPRi knockdown, something that is now facilitated by the generation of HaemoBrowse (Fig 7). Moreover, the pooled growth approach described here is robust and convenient, but unable to differentiate between competitive and environmental influences on sgRNA selection, as sgRNA selection in the sample may be influenced when cells with different sgRNA content grow together. Thus, as it is shown in Figs 3 and 4, and in **Figs D and E in S1 Text**, it

is required to perform confirmation experiments. Notably, our confirmation assays strongly validate the robustness and reliability of the CRISPRi-seq platform presented in this study. It should also be noted that we observed an outgrowth around 8–10 h for some individual knockdown mutants. This phenomenon was previously reported [43,44] and may be merely explained by a degradation of aTc or a mutation in the *tetR* gene or the P$_{tet}$. We cannot rule out partial repressions for which proteins might accumulate in the cytoplasm over time reaching a level that promotes a net growth. However, given that our CRISPRi screens were performed over maximum 21 generations (~10 h), this regrowth effect should have a minor impact on the data analysis output.

Importantly, our differential essentialome analysis showcased medium-dependent essential gene specificities. Depletion of many genes involved in purine (*guaB*) and amino acid (e.g. *ilvE* and *serA*) metabolism showed a marked impact on growth in CDM when compared to sBHI. The nutrient scarcity in CDM is supposed to be the main reason for such requirements. Notably, the system may also have the potential to titrate metabolite concentration requirements, as sgRNA$_{serA}$ abundance drops in a medium containing about 10 times less molecules of L-serine (CDM). Likewise, knockdown defect restoration is feasible, as seen upon addition of exogenous extra L-serine in CDM assays. In contrast, more puzzling results were found for genes with stronger impact on growth in sBHI compared to CDM. Knocking down genes encoding ABC transporter systems for metal (Zn, *zevB*; Mo, *modA*; or Ni, HI1624) or carbon sources (ribose, *rbsB*; dipeptides, *dpp* genes; methionine, *metQ*; polyamines, *potBC*) in a complex and rich medium such as sBHI is toxic. This might be due to a defect in nutrient import, or in pumping out nutrients from cytoplasm that could provoke an accumulation of toxic intermediates. These observations together reinforce the versatility of our CRISPRi platform, and open a whole range of clinically relevant *in vitro* or *in vivo* conditions to be investigated in future studies. Indeed, our CRISPRi platform is likely to be exploitable to a large set of clinically relevant *H. influenzae* strains or lineages to elicit a core set of essential genes. In this scope, our sgRNA library has already been designed to be compatible with three additional *H. influenzae* strains, NTHi375, 86–028NP and R2866 (see Material & Methods section). Moreover, this proof of concept study paves the way to multi-species experiments (genes beneficial/detrimental in bacterial co-cultures) and to host-pathogen interaction works where gene knockdown fitness will be evaluated in colonization, carriage, virulence or infectious conditions. In summary, the genetic (inducible promoter, neutral insertion platform, depletion system, genome-wide fitness library) and *in silico* (HaemoBrowse) resources generated here provide powerful tools to deeply investigate the molecular genetics of *H. influenzae*, facilitating the understanding of this pathogen physiology and the identification of therapeutic targets.

## Materials and methods

### Bacterial strains, growth conditions and chemicals

Strains used in this study are listed in **Table A in** S1 Text. *H. influenzae* strains were grown on solid medium at 37°C with 5% CO$_2$ using either PolyVitex agar (PVX, bioMérieux, 43101), or *Haemophilus* Test Medium agar (HTM, Oxoid, CM0898) supplemented with 10 µg/mL hemin and 10 µg/mL nicotinamide adenine dinucleotide (NAD) (sHTM). *H. influenzae* liquid cultures were grown at 37°C with 5% CO$_2$ in either brain heart infusion medium (BHI, Oxoid, CM1135), chemically defined medium (CDM) or minimal chemically defined medium (mCDM, see **Table B in** S1 Text), in all cases supplemented with 10 µg/mL hemin and 10 µg/mL NAD (referred to sBHI, CDM or mCDM). Stock solutions of anhydrotetracycline 500 µg/mL (aTc, Merck, 37919–100MG-R) were prepared in ethanol 96%. L-Serine (Merck, S4500) 0.47 M stock solution was prepared in distilled water. For *H. influenzae*, erythromycin 11 µg/mL (Erm$_{11}$), spectinomycin 50 µg/mL (Spec$_{50}$), or L-serine 10 mM were used when necessary. *E. coli* was grown on Luria Bertani (LB) or LB agar at 37°C, supplemented with Spec$_{50}$ when appropriate.

### Construction of a *H. influenzae* host strain with an aTc-inducible dCas9

Plasmids and primers used in this work are shown in **Tables C and D in** S1 Text, respectively. The aTc-inducible system to control *dcas9* expression was based on a previously developed doxycycline-inducible CRISPRi system in *S. pneumoniae* [19]. Four fragments (1–4) were amplified separately for assembly using Golden Gate cloning. Fragment 1,

containing the *H. influenzae xylB* region for homologous recombination, was amplified from RdKW20 genomic DNA with F_*xylB*/2469 and R_*Aar*I_*xylB*-ery/2470 primers; fragment 2, containing an Erm (*ermC*) selection marker, was amplified from pUC19-Hi061.1-Erm with F_*Aar*I_ery/2471 and R_*Aar*I_ery/2472 primers; fragment 3, containing the *tetR*-P$_{tet}$-*dcas9* cassette, was amplified from *S. pneumoniae* VL3468 strain genomic DNA [19] with F_*Aar*I_*tetR*/2473 and R_*Aar*I_dCas9/2474 primers; and fragment 4, containing the *H. influenzae rfaD* region for homologous recombination, was amplified from RdKW20 genomic DNA with F_*Aar*I_*rfaD*/2475 and R_*rfaD*/2476 primers. The four DNA fragments were separately purified using NucleoSpin Gel and PCR Clean-up Kit (Macherey-Nagel, 22740609.250), and assembled by Golden Gate cloning to build the P$_{tet}$-*dcas9* cassette. The assembly reaction, performed with *Aar*I (New England Biolabs, ER1581) and T4 DNA ligase (New England Biolabs, M0202S), involved 30 cycles of 1.5 min/cycle at 37ºC, followed by 3 min at 16ºC. Enzymes were inactivated at 80ºC for 10 min. The 8,424 bp ligation product was amplified by PCR with primers F_*xylB*/2469 and R_*rfaD*/2476, purified using NucleoSpin Gel and PCR Clean-up Kit, and chromosomally integrated into the RdKW20 genome by double homologous recombination between the *xylB* and *rfaD* genes using the M-IV method [45]. Chromosomal integration was selected on sHTM-agar with Erm$_{11}$, to generate the RdKW20 derivative strain RdKW20-*dcas9*, used as a host recipient strain for individual sgRNA and for sgRNA library cloning (named as *dcas9* strain).

## Generation of a vector platform for sgRNA cloning

Five fragments were assembled to generate the backbone vector pPEPZHi-*mCherry*, a derivative of pPEPZ-sgRNAclone [16,19]. Fragment 1, containing the pUC18 replication origin, was amplified from pPEPZ-sgRNAclone vector with F_*Aar*I-oUC18/2483 and R_*Aar*I-oUC18/2484 primers; fragment 2, containing the *H. influenzae* Hi0601.1 upstream region for homologous recombination, was amplified from RdKW20 genomic DNA with F_*Aar*I-Hi0601up/2477 and R_*Aar*I-Hi0601up/2478 primers; fragment 3 containing Illumina read 1 sequence, P3 promoter, *mCherry*-encoding gene flanked by *Bsm*BI restriction sites, dCas9 handle binding, terminator region of sgRNA, Illumina read 2 sequence, 8 bp Illumina index sequence and P7 adaptor sequence, was amplified from pPEPZ-sgRNAclone with primers F_*Aar*I-P3-sgRNA/2488 and R_*Aar*I-sgRNA/2480; fragment 4, containing the 5' end of a Spec$^R$ gene (nucleotides from 1 to 1,333) was amplified from pUC19-Hi061.1-Spec [46] with F_*Aar*I-specHi/2485 and R_*Aar*I_specmut/2517 primers; fragment 5 containing the rest of the same Spec$^R$ marker (nucleotides from 1,334–1,948) and the *H. influenzae* Hi0601.1 *locus* downstream region for homologous recombination, was amplified from pUC19-Hi061.1-Spec with F_*Aar*I_specmut/2518 and R_*Aar*I_Hi0601dn/2482 primers. The five fragments were purified separately using NucleoSpin Gel and PCR Clean-up Kit, and used in a Golden Gate assembly reaction with *Aar*I and T4 ligase, for 30 cycles of 1.5 min at 37ºC followed by 3 min at 16ºC. Enzymes were inactivated at 80ºC for 10 min. The Golden Gate product, i.e., backbone vector, was directly used to transform chemically competent *E. coli* Stbl3. Growth of red Spec$^R$ colonies after plating indicated successful transformation of *E. coli* Stbl3 with pPEPZHi-*mCherry*.

## Construction of individual sgRNA plasmids

Targeting sequences were ordered from Stab Vida (https://www.stabvida.com/es) as two partially complementary 24 nt primers (**Table D in** S1 Text). Annealing of each oligo pair generated a dsDNA fragment with 20 nt sgRNA base-pairing spacer sequence and 4 nt overhang at each end. The annealing reaction was performed separately for each sgRNA in 10xTEN buffer (10 mM Tris, 1 mM EDTA, 100 mM NaCl, pH 8) in a thermocycler, for 5 min at 95ºC, followed by slow cooldown at RT (0.1ºC/s). The two 4 nt overhangs were designed to be compatible with the two adhesive ends of the *Bsm*BI-digested pPEPZHi-*mCherry*. The pPEPZHi-*mCherry* vector was *Bsm*BI-digested (Fisher Scientific, ER0451) and purified by gel extraction to ensure removal of the 739 bp *mCherry* gene. Each annealing product was individually ligated into *Bsm*BI-digested pPEPZHi-*mCherry* with T4 ligase to generate pPEPZHi-sgRNA*geneX*clone plasmids. Ligation products were transformed into chemically competent *E. coli* Top10 cells. Transformants were selected on LB agar with Spec$_{50}$. After red/white screening and PCR confirmation, one colony from each pPEPZHi-sgRNAclone was selected for culturing and plasmid purification using PureYield Plasmid Miniprep System

(Promega, A1222). Purified plasmids were used to amplify each sgRNA cassette by PCR with F_AarI-Hi0601up/2477 and R_AarI_Hi0601dn/2482 primers. PCR products were transformed into the *dcas9* strain, generating RdKW20-*dcas9* P3-sgRNA$_{geneX}$. Three individual colonies per construct were stored and used for later analyses.

## Growth assay of *H. influenzae* CRISPRi system with individual sgRNAs

*H. influenzae dcas9* P3-sgRNA$_{geneX}$ clones were grown on PVX agar for 12 h at 37ºC with 5% $CO_2$. Two to five colonies were inoculated in 10 mL sBHI or CDM and grown for 12 h with shaking (90 r.p.m). Cultures were then diluted to $OD_{600} = 0.05$, incubated in sterile 50 mL flasks with 10 mL sBHI or CDM and shaking (180 r.p.m), up to $OD_{600} = 0.3$. Cultures were diluted for a second time to $OD_{600} = 0.03$, each in duplicate, incubated in sterile 50 mL flasks with 10 mL sBHI or CDM, in the presence or absence of aTc 50 ng/mL for 2 h at 180 r.p.m, up to $OD_{600} = 0.3$. Afterwards, cultures were diluted to $OD_{600} = 0.01$ in sBHI or CDM, and 198 µL-aliquots were transferred to individual wells in 96-well plates, in the respective presence or absence of aTc 50 ng/mL. Plates were incubated at 37ºC for 15 h in an Agilent Biotek Synergy H1 Microplate Reader H1M Unit3 – AV; $OD_{600}$ was recorded every 30 min. For the $P_{tet}$ responsiveness experiment, cells were grown for 12 h at 37°C in 10 mL sBHI with shaking (120 r.p.m). Next, cultures were diluted in fresh sBHI to $OD_{600} = 0.01$ and transferred to a 96-wells plate with a range of aTc concentrations. In all cases, each growth curve was corrected to its respective blank value. All 96-well experiments were performed in duplicate on at least three independent occasions (n ≥ 3).

## sgRNA library design

Prokka [47] was used to annotate the *H. influenzae* genomes of strains RdKW20, NTHi375, 86–028NP and R2866, and the non-redundant sgRNA library was automatically designed via a R-pipeline (https://github.com/veeninglab/CRISPRi-seq) to target every genetic feature as previously described [16,48]. The list of 7,260 sgRNAs designed for the 4 aforementioned strains was narrowed down to generate a non-redundant library of 3,351 sgRNAs (1,773 sgRNAs specific to strain RdKW20) and avoid exact duplicates that will bias the library distribution. Detailed information about the list of all sgRNA sequences and target genes is available in S7 Dataset.

## Construction of a genome-wide sgRNA plasmid pool

An oligo pair was designed for each of 3,351 sgRNAs, and 6,702 oligonucleotides were manufactured by IDT (S7 Dataset), corresponding to two partially anti-complementary 24 nt oligos per annotated ORF at a final concentration of 1 pmol/oligo (oPools). The annealing process of the oligo pairs in pool resulted in the formation of a double stranded DNA fragment with 20 nt sgRNA base-pairing spacer sequence with a 4 nt 5′ overhang, as described above. Next, the annealing product was 5′ phosphorylated with T4 polynucleotide kinase (New England BioLabs, M0201S) for 40 min at 37ºC, and inactivated by incubating at 65ºC for 20 min. The two 4 nt overhangs were designed to align with the two adhesive ends of the *Bsm*BI-digested pPEPZHi-*mCherry*. To guarantee *mCherry* elimination, a digestion was performed with the *Rrs*II restriction enzyme, which has a cutting site within the *mCherry* gene. The *Rrs*II digestion product was used as a template for PCR with primers OVL6185 and OVL6186, and purified using NucleoSpin Gel and PCR Clean-up Kit. The purified PCR product was digested with *Bsm*BI and ligated to the phosphorylated pool-annealing product in a one step process performed in a thermocycler, for 1.5 min at 37ºC, 66 cycles of 3 min at 16ºC and 5 min at 37ºC, followed by 10 min at 80ºC, to generate a pPEPZHi-sgRNAlibrary plasmid pool. The pPEPZHi-sgRNAlibrary plasmid pool was transformed into electrocompetent *E. coli* Stbl3 cells. Spec$^R$ transformants were pooled, collected and stored at -80ºC in LB-glycerol 20%. Collected transformant pools were used for plasmid pool mini-prep purification.

## Construction of CRISPRi library in *H. influenzae*

The purified pPEPZHi-sgRNAlibrary plasmid pool was used to generate a genome-wide CRISPRi library in the *dcas9* strain. The purified sgRNA plasmid pool was separately digested with (i) *Hae*II (New England BioLabs, R0107S) and *Rsr*II (New England BioLabs, R0501S); (ii) *Mme*I (New England BioLabs, R0637S) and *Rsr*II, for 1 h at 37ºC, and restriction

enzymes were further inactivated for 20 min at 80ºC or 65ºC, respectively. This generates a pool of linear DNA fragments, mixed together in equal proportion (~60 $\mu$L of the digestion product), and used for transformation of naturally competent *dcas9* bacterial cells using the M-IV method. After the incubation period, the entire volume was plated on HTM-agar with Spec$_{50}$ (300 $\mu$L/plate on 150 mm petri dishes), and plates were incubated at 37ºC overnight with 5% $CO_2$. Pooled transformants were collected with 5 mL sBHI and stored at -80ºC in sBHI-glycerol 20%.

## CRISPRi-seq essentiality screen

The *dcas9* CRISPRi library was grown in 25 mL fresh sBHI or CDM medium using 500 $\mu$L aliquots of the previously frozen CRISPRi library. Cultures were grown at 37ºC with shaking (180 r.p.m) to OD$_{600}$ = 0.3, and stored (2.5 mL culture + 750 $\mu$L 80% glycerol) at -80ºC, named working stocks. Library growth for CRISPRi-seq assay was performed using the working stocks. When appropriate, 50 ng/mL aTc was added for induction. All conditions were performed in quadruplicate. For the 7 generations experiments, 250 $\mu$L of working stock were inoculated in 24.75 mL medium in the absence (control group) or presence (experimental group) of aTc 50 ng/mL. Cultures were incubated at 37ºC with shaking (180 r.p.m) to OD$_{600}$ = 0.3. Ten mL/culture were next centrifuged (4,000 g, 10 min, 4ºC), and pellets were kept at -80ºC for subsequent genomic DNA extraction. For 14 generations experiments, the previous cultures were 1:100 diluted (250 $\mu$L culture in 24,75 mL medium), and incubated at 37ºC, 180 r.p.m. When OD$_{600}$ reached 0.3, 10 mL/culture were harvested by centrifugation (4,000 g, 10 min, 4ºC), and pellets were kept as above. For 21 generations assays, cultures from the 14 generations assay were 1:100 diluted as before, incubated at 37ºC and 180 r.p.m to reach OD$_{600}$ = 0.3, 10 mL/culture were centrifuged (4,000 g, 10 min, 4ºC), and pellets were kept at -80ºC.

## Library preparation and Illumina sequencing

Samples were used for genomic DNA extraction and purification with DNeasy Blood & Tissue kit (Qiagen, 69506), resuspended in 200 $\mu$L DNase-free water/sample, and quantified with a Nanodrop spectrophotometer. Genomic DNA samples served as a template for a one step PCR reaction with primers including barcode index (N5 and N7 Illumina series). PCR products were purified from gel extraction, quantified (Qubit dsDNA HS Assay Kit; life Technologies, Q32851) and assembled in equimolarity for library pooling deep sequencing. Purified amplicons were sequenced on an Aviti systems (Element Biosciences) using single ends 150 bp reads protocols and read primer 1. Sequencing data are available as a BioProject, ID: PRJNA1274682.

## CRISPRi-seq differential enrichment analysis

Sequencing results were demultiplexed and processed through 2FAST2Q (available at https://github.com/veeninglab/2FAST2Q) to elicit sgRNA counts per sample. Differential enrichment analysis using the R package DESeq2 was performed for evaluation of fitness cost of each sgRNA, as previously described. EggNOG mapper v2 (http://eggnog-mapper.embl.de/) [49] was used to allocate COG categories (one letter code) to each genetic feature of the prokka annotated genome of RdKW20. Enrichment pathway analyses were performed with the R packages ClusterProfiler and enrichlot. Metabolic maps were generated with the R package pathview.

## Generation of Δ*serA* mutant strain

 To inactivate the *serA* gene in the RdKW20 strain, a 1,240 bp DNA fragment corresponding to the *serA* coding sequence was PCR amplified (*Phusion* DNA Polymerase) using RdKW20 genomic DNA as template with primers f1_serA_new/2333 and r1_serA_new/2334. This amplicon was cloned into pJET1.2/blunt, generating pJET1.2-*serA*/P1259, which was then linearized by inverse PCR using primers serA_F2_NEW/2376 and serA_R2_NEW/2377 to disrupt the *serA* gene sequence, followed by ligation to a blunt-ended Spec$^R$ gene obtained from pRSM2832 by *EcoR*V digestion, to generate

pJET1.2-*serA*::*spec* (P1260). The *serA::spec* disruption cassette (2,410 bp) was PCR amplified with primers f1_serA_new/2333 and r1_serA_new/2334, and used for RdKW20 natural transformation using the M-IV method [45]. The Δ*ser-A::spec* (P1261) strain was selected on sHTM agar with $Spec_{50}$ and confirmed by PCR.

## Microscopy imaging and image processing

Cells were grown for 12 h in 50 mL-falcon tubes with 10 mL sBHI at 37°C with shaking (120 r.p.m). They were next diluted in fresh sBHI to $OD_{600}$ = 0.05 and incubated at 37°C with shaking (120 r.p.m) for 3 h, in the presence or absence of aTc. When stated, cells were stained with DAPI (Sigma Aldrich, D9542). Briefly, 500 $\mu$L of cells were incubated at 37°C with DAPI 100 $\mu$g/mL for 15 min, and next centrifuged (5,000 r.p.m. for 5 min) and washed with 1 mL sBHI. Cells were imaged by adding 1 $\mu$L of cell suspension onto 1% (w/v) PBS-agarose gel pads. Microscopy was conducted using a Leica DMi8 microscope with a sCMOS DFC9000 GTC (Leica) camera and a ×100/1.40 oil-immersion objective (0.09 working distance). Phase-contrast images were acquired using transmission light with a 100 ms exposure. Snapshot fluorescence images were acquired with a 100 ms exposure, a 405 nm excitation laser module and a DAPI 390 filter (Ex: 395/25 nm, BS: LP 425 Leica 11533333, Em: 460/50 nm). All microscope images were acquired with LasX v.3.4.2.18368 (Leica), and processed with FIJI v2.14.0 to adjust contrast and brightness [50].

## HaemoBrowse

The HaemoBrowse genome browser for *H. influenzae* was created using JBrowse 2 [51], through methods described before [52]. In short, the genomes of 86–028NP, NTHi375, R2866, RdKW20 (Refseq accession numbers GCF_000012185.1, GCF_000767075.1, GCF_000165525.1, and GCF_000027305.1, respectively) were acquired from the NCBI, and annotated *de novo* using Prokka [46]. The protein sequences were used to generate links to UniProt [37], PaperBLAST [38], AlphaFold [39], FoldSeek [40] and STRING [41] for each protein coding sequence, by matching through BLAST [53]. Translated protein sequences from strain RdKW20 were also used to match the Prokka annotations to the reference locus tags for RdKW20 (HI_XXXX). Prokka annotation information was used for links to external databases on enzyme classifiers (Enzyme Commission (EC) numbers, and Clusters of Orthologous Groups (COGs)). The designed sgRNAs targeting each genome are available from each genome's sgRNA track. Conservation was determined using panaroo [54] from the Prokka-produced gff-files, through standard settings and "clean-mode" set to "strict". Genome alignment was determined using MashMap3 (v3.1.3) [55]. The NucContent plugin for JBrowse2 is used to provide a track to calculate and visualize GC content (available from https://github.com/jjrozewicki/jbrowse2-plugin-nuccontent).

## Supporting information

**S1 Dataset. sgRNA genome-wide library for *H. influenzae* RdKW20.**
(XLSX)

**S2 Dataset. CRISPRi library distribution in *E. coli* Stbl3 and *H. influenzae* RdKW20.**
(XLSX)

**S3 Dataset. *H. influenzae* essential genetic features upon growth in sBHI or in CDM, for 7, 14 and 21 generations.**
(XLSX)

**S4 Dataset. *H. influenzae* differentially essential genetic features upon growth in sBHI or CDM.**
(XLSX)

**S5 Dataset. Functional categories of *H. influenzae* differentially essential genes upon growth in sBHI or CDM.**
(XLSX)

**S6 Dataset. Comparison of RdKW20 essential genes identified in CRISPRi-seq or via transposon-mutant libraries.**
(XLSX)

**S7 Dataset. sgRNA genome-wide library for *H. influenzae* strains RdKW20, 86–028NP, NTHi375 and R2866.**
(XLSX)

**S1 Text. Supplementary figures and tables: Table A.** Bacterial strains used in this study. **Table B.** Composition of CDM and mCDM media used in this study. **Table C.** Plasmids used in this study. **Table D.** Primers used in this study. **Fig A.** Dynamic range of the aTc-inducible CRISPRi system in *H. influenzae*. (A) Growth of *H. influenzae* WT (triangles) and *dcas9* (circles) strains in sBHI. (B) $P_{tet}$ responsiveness experiment: *dcas9* strains carrying no (-) or sgRNAs targeting the *sspA*, *fabH* or *glyA* genes were grown for 12 h at 37°C in 10 mL sBHI with shaking (120 r.p.m). Next, cultures were diluted in fresh sBHI to $OD_{600} = 0.01$ and transferred to a 96-wells plate with a range of aTc concentrations from 0.25 to 50 ng/mL. The color-coded shadow area represents the standard deviation to the mean (solid line). **Fig B.** Frequency histogram of the *E. coli* and *H. influenzae* sgRNA library distribution. sgRNA library distribution in *E. coli* (A) and in *H. influenzae* (B). **Fig C.** sgRNA content variation during CRISPRi screen in sBHI and CDM. (A) Principal component analysis (PCA) shows the disparity between replicates, induction (+ or – aTc), media (sBHI, CDM) and timepoints (7, 14 and 21 generations). + signs represent samples where CRISPRi was activated by aTc; - signs show samples where CRISPRi was uninduced. Blue and red signs represent CRISPRi libraries grown in sBHI and CDM, respectively. Increasing the induction time explains most of the variance in sample strain composition (91%). (B) Clustered heatmap of sgRNA content between replicates, induction (+ or – aTc), media (sBHI, CDM) and timepoints (7, 14 and 21 generations). The normalized read count is expressed in a gradient logarithmic scale. The horizontal axis displays the 1,773 sgRNAs designed for the RdKW20 CRISPRi library, while the vertical axis shows the 48 different samples. **Fig D.** CRISPRi-based analysis of *H. influenzae* specific gene fitness in sBHI. *dcas9* derivative strains expressing *infC*, *rpsL*, *ispE*, *metK*, *rpoC*, *lpxB*, *parE*, *rsxA*, *rpsT* and *mepA* sgRNAs were grown in sBHI, in the absence (white circles)/presence (black circles) of aTc. Strains were grown in 96-well plates, $OD_{600}$ was measured every 30 min for 15 h; standard deviation to the mean is shown for each timepoint. **Fig E.** CRISPRi-based analysis of *H. influenzae* specific gene fitness in CDM. *dcas9* derivative strains expressing *infC*, *rpsL*, *ispE*, *metK*, *rpoC*, *lpxB*, *parE*, and *rsxA*, *rpsT* and *mepA* sgRNAs were grown in CDM, in the absence (white circles)/presence (black circles) of aTc. Strains were grown in 96-well plates, $OD_{600}$ was measured every 30 min for 15 h; standard deviation to the mean is shown for each timepoint. **Fig F.** Pathway enrichment in sBHI. KEGG pathway enrichment analysis in sBHI (*versus* CDM). The gene ratio refers to the number of enriched genes in sBHI for a specific pathway reported to the total amount of differentially essential genes. Dot size reflects the number of differentially essential genes. Color shades scale the significance (Padj). **Fig G.** Metabolic map of differentially essential gene enrichment in sBHI. KEGG map (hin01100) of the metabolic landscape in *H. influenzae* RdKW20 (green). The nodes represent metabolic compounds, and edges represent genes involved in the reactions. Pathways with enrichment in differentially essential genes are framed in orange. Metabolic enzymes differentially more essential in sBHI (*versus* CDM) are highlighted in red. **Fig H.** Metabolic map of differentially essential gene enrichment in CDM. KEGG map of the metabolic landscape in *H. influenzae* RdKW20 (green). The nodes represent metabolic compounds and edges represent genes involved in the reactions. Pathways with enrichment in differentially essential genes are framed in orange. Metabolic enzymes differentially more essential in CDM (*versus* sBHI) are highlighted in red. **Fig I.** Differentially essential genes in the glycine, serine and threonine pathway. Path view mapper visualization of enriched/depleted genes using the KEGG database (hin00260) in CDM *versus* sBHI for the glycine, serine and threonine metabolism. KEGG reference pathway numbers are replaced by gene names when homologs are encoded in the RdKW20 genome. The $log_2FC$ of fitness scores (from DEseq2 analysis) are depicted with a gradient color scale. Red color indicates a negative $log_2FC$. **Fig J.** sgRNA gene silencing reversibility in *H. influenzae*. (A) The *dcas9* sgRNA$_{serA}$ strain was grown in mCDM (left) or mCDM + L-serine

10 mM (right), in the absence (white circles)/presence (black circles) of aTc. (B) RdKW20 WT and Δ*serA* mutant strains were grown in mCDM in the absence (white circles)/presence (grey circles) of L-serine 10 mM.
(DOCX)

## Acknowledgments

We thank Drs. Begoña Euba, Nahikari López-López and Javier Asensio-López for helping to define mCDM and L-serine specificities needed for this work.

## Author contributions

**Conceptualization:** Celia Gil-Campillo, Johann Mignolet, Axel B. Janssen, Vincent de Bakker, Jan-Willem Veening, Junkal Garmendia.

**Data curation:** Johann Mignolet.

**Formal analysis:** Celia Gil-Campillo, Johann Mignolet, Asier Domínguez-San Pedro, Junkal Garmendia.

**Funding acquisition:** Jan-Willem Veening, Junkal Garmendia.

**Investigation:** Celia Gil-Campillo, Johann Mignolet, Axel B. Janssen, Vincent de Bakker, Junkal Garmendia.

**Methodology:** Celia Gil-Campillo, Johann Mignolet, Asier Domínguez-San Pedro, Beatriz Rapún-Araiz, Axel B. Janssen, Vincent de Bakker.

**Project administration:** Jan-Willem Veening, Junkal Garmendia.

**Resources:** Jan-Willem Veening, Junkal Garmendia.

**Supervision:** Johann Mignolet, Jan-Willem Veening, Junkal Garmendia.

**Writing – original draft:** Celia Gil-Campillo, Johann Mignolet, Axel B. Janssen, Junkal Garmendia.

**Writing – review & editing:** Celia Gil-Campillo, Johann Mignolet, Asier Domínguez-San Pedro, Beatriz Rapún-Araiz, Axel B. Janssen, Vincent de Bakker, Jan-Willem Veening, Junkal Garmendia.

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
