## [Decision Letter · Decision Letter 0]

28 Sep 2025

PPATHOGENS-D-25-01952

CRISPRi-seq in Haemophilus influenzae reveals genome-wide and medium-specific growth determinants

PLOS Pathogens

Dear Dr. Garmendia,

Thank you for submitting your manuscript to PLOS Pathogens. Your manuscript was evaluated by member so the editorial board and three external referees. All were enthusiastic about your study but had some concerns (see below). Therefore, we invite you to submit a revised version of the manuscript that addresses all of the points raised during the review process, especially include the gene lists, and appropriate discussion, as noted by Reviewer 3. 

Please submit your revised manuscript within 30 days Nov 23 2025 11:59PM. If you will need more time than this to complete your revisions, please reply to this message or contact the journal office at plospathogens@plos.org. Please include the following items when submitting your revised manuscript:

We look forward to receiving your revised manuscript.

Kind regards,

D. Ashley Robinson, Ph.D.

Academic Editor

PLOS Pathogens

D. Scott Samuels

Section Editor

PLOS Pathogens

Sumita Bhaduri-McIntosh

Editor-in-Chief

PLOS Pathogens

orcid.org/0000-0003-2946-9497

Michael Malim

Editor-in-Chief

PLOS Pathogens

orcid.org/0000-0002-7699-2064

**Journal Requirements:**

1) We do not publish any copyright or trademark symbols that usually accompany proprietary names, eg ©,  ®, or TM  (e.g. next to drug or reagent names). Therefore please remove all instances of trademark/copyright symbols throughout the text, including:

- ® on page: 23

- TM on page: 21.

2) We have noticed that you have uploaded Supporting Information files, but you have not included a list of legends. Please add a full list of legends for your Supporting Information files after the references list.

3) Some material included in your submission may be copyrighted. According to PLOSu2019s copyright policy, authors who use figures or other material (e.g., graphics, clipart, maps) from another author or copyright holder must demonstrate or obtain permission to publish this material under the Creative Commons Attribution 4.0 International (CC BY 4.0) License used by PLOS journals. Please closely review the details of PLOSu2019s copyright requirements here: PLOS Licenses and Copyright. If you need to request permissions from a copyright holder, you may use PLOS's Copyright Content Permission form.

Potential Copyright Issues:

i) Figures 1C, and 2A. Please confirm whether you drew the images / clip-art within the figure panels by hand. If you did not draw the images, please provide (a) a link to the source of the images or icons and their license / terms of use; or (b) written permission from the copyright holder to publish the images or icons under our CC BY 4.0 license. Alternatively, you may replace the images with open source alternatives. See these open source resources you may use to replace images / clip-art:

ii) The following Figure contains screenshots: 7D. We are not permitted to publish these under our CC-BY 4.0 license, websites are usually intellectual property and are copyrighted.This includes peripheral graphics of the web browser such as icons and button. We ask that you please remove or replace it.

4) Thank you for stating "Data available as a BioProject, ID: PRJNA1274682." Please note that, though access restrictions are acceptable now, your entire minimal dataset will need to be made freely accessible if your manuscript is accepted for publication. This policy applies to all data except where public deposition would breach compliance with the protocol approved by your research ethics board. 

5) Thank you for indicating that “HaemoBrowse is freely available at https://HaemoBrowse.VeeningLab.com.” Thank you for uploading your study's underlying data set. Unfortunately, the repository you have noted in your Data Availability statement does not qualify as an acceptable data repository according to PLOS's standards. At this time, please upload the minimal data set necessary to replicate your study's findings to a stable, public repository (such as figshare or Dryad) and provide us with the relevant URLs, DOIs, or accession numbers that may be used to access these data. For a list of recommended repositories and additional information on PLOS standards for data deposition, please see

https://journals.plos.org/plospathogens/s/recommended-repositories

7) Thank you for stating "J.W.V. is a scientific advisory board member at i-Seq Biotechnology." Please declare all competing interests beginning with the statement "I have read the journal's policy and the authors of this manuscript have the following competing interests:"

8) Please ensure that the funders and grant numbers match between the Financial Disclosure field and the Funding Information tab in your submission form. Note that the funders must be provided in the same order in both places as well. 

**Reviewers' Comments:**

Reviewer's Responses to Questions

**Part I - Summary**

Reviewer #1: This paper presents a fascinating development in functional genomics to define essential genes and explore fitness impacts across the entire H. influenza genome. There are current limitations (most notably use of model strain Rd which is not fully representative of either encapsulated or nonencapsulated lineages in carriage or disease), and the conditions for which fitness is tested are fairly limited...reviewer would certainly like to see this work extended into more relevant strains/lineages and conditions associated with carriage or virulence. These points certainly don't limit the significance or impact of the work in this paper

Reviewer #2: This manuscript by Gil-Campillo et al. describes the implementation and validation of an inducible CRISPR system for mutant generation in Haemophilus influenzae. The manuscript is well written and presented in a clear, logical manner that is well supported by the data. The approach is sound and strengthened by the use of varied methodology and phenotypic complementation. While the CRISPRi approach has been described in other bacteria, there is novelty in its application to Haemophilus as well as in the development of the HaemoBrowse database generated in this work. The validation of identified genes through direct mutagenesis, complementation, and phenotypic assays is a significant strength of the study.

Reviewer #3: PloS review Sept.23, 2025

This is a commendable study in which the authors report a comprehensive functional genomics tool for H. influenzae strain Rd KW20. While primarily a methods paper, the authors demonstrate cell and physiological phenotypes, such as cell division and nutrient utilization, that can be studied using the sgRNA conditional expression system. Overall, the findings are straightforward and confirm or extend results of previous essential gene identifications provided by transposon insertion site sequencing studies and demonstrate utility of an inducible genetic repression system that allows detailed confirmation and follow-up functional analyses of genes involved in fitness under varied conditions.

It would be useful to know the approximate percentage of sgRNA constructs that are functionally able to deplete their target RNAs.

From the standpoint of studying H. influenzae as a pathogen, a missed opportunity was the lack of an animal infection study, as the tool could potentially clarify which of the in vitro essential genes are also essential for infection.

Line 180: The term 'genetic features' is vague. Describe the type of elements that were targeted by the design. (eg. all protein coding genes, RNA genes, +?).

Lines 319-322: Some studies conducting transposon-based essential gene identification in H. influenzae were omitted and could improve the comparision. The comparison of the identified essential gene set with those generated in transposon studies should be reported directly as a list, with some discussion of genes that were discordant between methods. Also, were there any genes identified that were not found in transposon-based studies?

Lines 337-338: This sentence confounds an experimental caveat (mixed competition) with statistical significance.

Line 348: The term "showcased medium specificities" is ambiguous. Do the authors mean differences in media composition? If so, this is circular logic as we already know the media compositions are different, and the main constituents are known, so it is not clear why it would need to be showcased.

Lines 355-359: The phrase "differentially essential gene enrichment in sBHI" should be clearly defined. The specific genes under discussion should be named and citations provided for the functions ascribed to these genes. The data supporting this discussion should be included in the manuscript so that readers do not need to access the online portal to interpret the statements.

Line 356: The authors should comment on whether the insertion of all of the sgRNA constructs into the xylose locus may have influenced essentiality results seen with xylose transport genes.

**Part II – Major Issues: Key Experiments Required for Acceptance**

Reviewer #1: As noted, ideally this would be extended to relevant strains or conditions, including in vitro or in vivo infection models. I think it's fair for that to be mentioned in discussion as an important future goal

Reviewer #2: None

Reviewer #3: (No Response)

**Part III – Minor Issues: Editorial and Data Presentation Modifications**

Reviewer #1: No Concerns

Reviewer #2: For many of the induced mutants, outgrowth is observed around 8–10 hours under several assay/growth conditions. This observation should be more directly addressed in the Discussion. Is this due to loss of induction or the emergence of suppressor mutants? Some discussion or a working hypothesis should be provided.

Throughout, “MI-V” should be changed to “M-IV” for consistency with the original cited publication.

Line 296: remove the repeated word “designed”

Reviewer #3: (No Response)

PLOS authors have the option to publish the peer review history of their article (what does this mean? ). If published, this will include your full peer review and any attached files.

**Do you want your identity to be public for this peer review?** For information about this choice, including consent withdrawal, please see our Privacy Policy .

Reviewer #1: No

Reviewer #2: No

Reviewer #3: No

**Figure resubmission:**

**Reproducibility:**



---

## [Editor Report · Decision Letter 1]

21 Oct 2025

Dear Dr. Garmendia,

We are pleased to inform you that your manuscript 'CRISPRi-seq in Haemophilus influenzae reveals genome-wide and medium-specific growth determinants' has been provisionally accepted for publication in PLOS Pathogens.

Best regards,

D. Ashley Robinson, Ph.D.

Academic Editor

PLOS Pathogens

D. Scott Samuels

Section Editor

PLOS Pathogens

Sumita Bhaduri-McIntosh

Editor-in-Chief

PLOS Pathogens

orcid.org/0000-0003-2946-9497

Michael Malim

Editor-in-Chief

PLOS Pathogens

orcid.org/0000-0002-7699-2064
---

## [Editor Report · Acceptance letter]

Dear Dr. Garmendia,

We are delighted to inform you that your manuscript, "CRISPRi-seq in Haemophilus influenzae reveals genome-wide and medium-specific growth determinants," has been formally accepted for publication in PLOS Pathogens.

Best regards,

Sumita Bhaduri-McIntosh

Editor-in-Chief

PLOS Pathogens

orcid.org/0000-0003-2946-9497

Michael Malim

Editor-in-Chief

PLOS Pathogens

orcid.org/0000-0002-7699-2064